# Structure of *Aedes aegypti* procarboxypeptidase B1 and its binding with Dengue virus for controlling infection

Edem Gavor[1], Yeu Khai Choong[1], Nikhil Kumar Tulsian[1,2], Digant Nayak[1], Fakhriedzwan Idris[3,4], Hariharan Sivaraman[1], Donald Heng Rong Ting[3,4], Alonso Sylvie[3,4], Yu Keung Mok[1], R Manjunatha Kini[1,5], J Sivaraman[1]

Metallocarboxypeptidases play critical roles in the development of mosquitoes and influence pathogen/parasite infection of the mosquito midgut. Here, we report the crystal structure of *Aedes aegypti* procarboxypeptidase B1 (PCPBAe1), characterized its substrate specificity and mechanism of binding to and inhibiting Dengue virus (DENV). We show that the activated PCPBAe1 (CPBAe1) hydrolyzes both Arg- and Lys-substrates, which is modulated by residues Asp[251] and Ser[239]. Notably, these residues are conserved in CPBs across mosquito species, possibly required for efficient digestion of basic dietary residues that are necessary for mosquito reproduction and development. Importantly, we characterized the interaction between PCPBAe1 and DENV envelope (E) protein, virus-like particles, and infectious virions. We identified residues Asp[18A], Glu[19A], Glu[85], Arg[87], and Arg[89] of PCPBAe1 are essential for interaction with DENV. PCPBAe1 maps to the dimeric interface of the E protein domains I/II (Lys[64]–Glu[84], Val[238]–Val[252], and Leu[278]–Leu[287]). Overall, our studies provide general insights into how the substrate-binding property of mosquito carboxypeptidases could be targeted to potentially control mosquito populations or proposes a mechanism by which PCPBAe1 binds to and inhibits DENV.

## Introduction

Mosquito-borne viruses such as dengue virus, Zika virus, and chikungunya virus, as well as the malaria parasite constitute a major threat to public health. The mosquito midgut serves as the first entry point for pathogen infection and replication (1, 2). The successful establishment of infection in the midgut cells is a prerequisite for pathogen survival and transmission. Studies have shown that some specific proteins in the midgut promote pathogenicity (3, 4, 5), whereas certain other proteins elicit antiviral/antiparasitic activities (6, 7).

In this regard, the midgut metallocarboxypeptidases (MCPs) have been considered as transmission-blocking vaccine candidates against mosquito and other insect vectors (4, 7, 8, 9, 10, 11). MCPs are exopeptidases involved in numerous processes, including digestion, blood coagulation/fibrinolysis, inflammation, and pro-hormone and neuropeptide processing (12). In mosquitoes, the A/B subfamily of MCPs (including carboxypeptidases A and B; CPB and CPA, respectively) is primarily responsible for dietary blood digestion (3, 13, 14) for the release of amino acid–rich nutrients required for egg development (4, 5, 15, 16). As such, MCPs are important targets for mosquito population control. MCPs also play a significant role in parasite transmission in mosquitoes. For example, the *Anopheles* midgut CPB1 has been shown to enhance malaria parasite transmission in various *Anopheles* mosquitoes (4, 5, 16). Active immunization with recombinant *Anopheles gambiae* procarboxypeptidase B1 (PCPBAg1) protected mice against *Plasmodium berghei* challenge (4). In addition, mosquitoes fed with anti-PCPBAg1 antibodies exhibited reduced reproductive capacity (5), a secondary effect of a CPB-based transmission-blocking vaccine that could likely suppress mosquito populations. This effect was also observed using inhibitors against mosquito dipeptidases. Similarly, the infection rate of *Plasmodium falciparum* gametocytes in mosquitoes was significantly reduced after treatment with anti-CPBAg1 antibodies (5, 16). Besides, inhibitors against peptidyl dipeptidase A administered to both *An. gambiae* and *Ae. aegypti* mosquito larvae led to stunted growth, reduced fecundity, and mortality (17).

In the midgut of *Ae. aegypti* mosquito, procarboxypeptidase B1 (CPB1 or PCPBAe1) takes on a different role in pathogen infection, interacting with the DENV-2 envelope (E) protein/virion and impeding viral packaging, maturation, and release from the midgut (6, 7). The interaction between PCPBAe1 and the E protein/virion can also inhibit viral morphogenesis and affect viral glycoprotein processing (7). The overall impact is the release of low titers of mature viruses. Moreover, another procarboxypeptidase A homolog in *Ae. aegypti* interacts with the DENV capsid protein (18); albeit, the

---

[1]Department of Biological Sciences, National University of Singapore, Singapore   [2]Department of Biochemistry, Yong Loo Lin School of Medicine, National University of Singapore, Singapore   [3]Infectious Diseases Translational Research Programme, Department of Microbiology and Immunology, Yong Loo Lin School of Medicine, National University of Singapore, Singapore   [4]Immunology Programme, Life Sciences Institute, National University of Singapore, Singapore   [5]Department of Pharmacology, Yong Loo Lin School of Medicine, National University of Singapore, Singapore

Correspondence: dbsjayar@nus.edu.sg

role of this interaction has yet to be established. Furthermore, the *Ae. aegypti* carboxypeptidase gene promoters drive the expression of genes that hinder parasite development in the guts of transgenic mosquitoes (19).

Despite the crucial roles of MCPs in mosquitoes, the structural and molecular mechanisms associated with these proteases are unclear. Here, we characterized the enzymatic activity of the mature peptidase domain (CPBAe1) and determined the structure of the full-length (FL)-PCPBAe1 (proenzyme) at 2.08-Å resolution. The results provide insights into the substrate specificity of this enzyme and show how this enzyme differs from other non-mosquito insects carboxypeptidases (11). We characterized the interactions between the recombinant PCPBAe1 and DENV-2 envelope (E)-protein, virus-like particles (VLPs), and infectious virions of different serotypes and mutants, and propose the mechanism of DENV suppression by PCPBAe1. Using hydrogen-deuterium exchange mass spectrometry (HDXMS), site-directed mutagenesis, and ELISA, we mapped the interaction interface between PCPBAe1 and the DENV-2 E protein. Collectively, our studies provide the opportunity (i) to design antibodies or inhibitors against mosquito carboxypeptidases and (ii) to develop novel strategies to control DENV propagation in the mosquito vector.

## Results

### Sequence analysis of the PCPBAe1 gene

Tham et al reported a putative procarboxypeptidase B1 (annotated *CPB1*), from *Ae. aegypti* mosquito midgut (6, 7). Here, we have adopted the conventional naming system to differentiate the full-length procarboxypeptidase (FL-*PCPBAe1*) from the mature peptidase domain alone (CPBAe1) with "Ae" denoting the *Ae. aegypti*'s

origin of the gene. Besides, the *PCPBAe1* amino acid residue numbering used here adopts the conventional numbering system of carboxypeptidases (10, 11, 20). Based on sequence analyses (Fig 1A), *PCPBAe1* is expressed as a preproprotein, with a signal peptide (Met*–Ala*), an N-terminal pro-region (Ala$^{1A}$–Asp$^{75A}$; "A" to indicate the activation unit or pro-region), and a C-terminal mature peptidase domain (Asp$^7$–Phe$^{305}$) (*CPBAe1*). There is an activation loop between Asp$^{75A}$ and Asp$^7$, with a primary trypsin cleavage site located between the scissile peptide bond, Arg$^6$-Asp$^7$ (Fig 1A). A unique feature of PCPBAe1 is the missing $3_{10}$-helix insertion, which is typical to CPBs (20, 21). From our analysis, we observe that this feature may only be unique to mammalian B-type MCPs and not insect B-type MCPs like *CPBAe1* and *CPBHz* (Fig 1B).

### Structure of PCPBAe1 and the pro-region auto-inhibition mechanism

Crystal structure of PCPBAe1 was determined to 2.08-Å resolution (Fig 2) 7EQX. PCPBAe1 elutes as a monomer in solution as revealed by size-exclusion chromatography and dynamic light scattering (Fig S1A–C). However, the structure consists of two molecules in the asymmetric unit of the crystal. Each molecule consists of 394 amino acid residues (Glu$^{5A}$–Phe$^{305}$) (Fig 2). The activation loop (Asp$^{75A}$–Asp$^7$) that connects the pro-region to the mature peptidase domain is disordered and could not be modeled in our structure. The pro-region adopts an open-sandwich antiparallel-*α*/antiparallel-*β*-fold, consisting of two *α*-helices and four *β*-strands. The mature peptidase domain contains a zinc-coordinated active site triad and the substrate-binding pocket and displays a core of eight twisted *β*-sheets surrounded by eight *α*-helices (Fig 2).

The globular N-terminal pro-region occupies the wide active site cleft of the mature peptidase domain, mediated via interactions with residues from the substrate-binding pocket of the peptidase

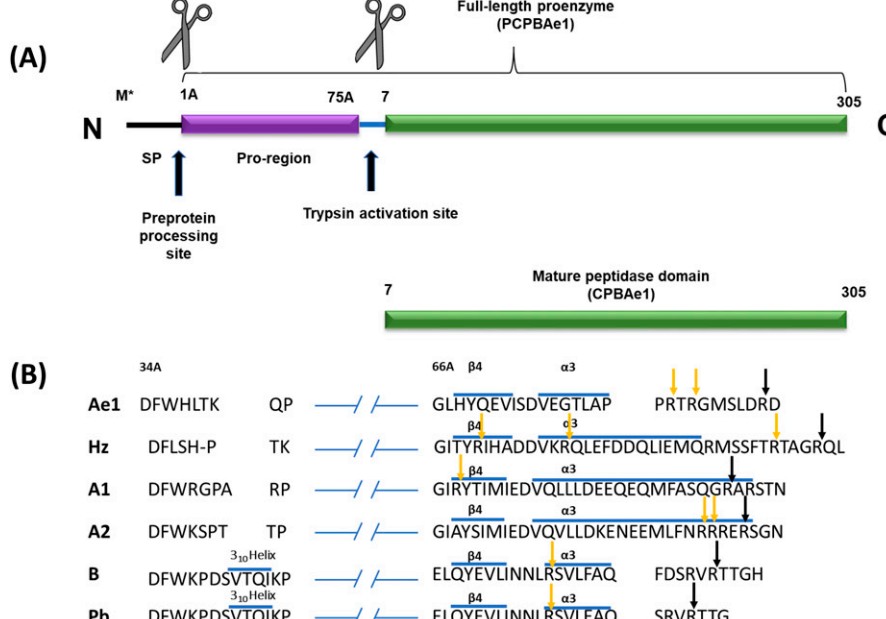

**Figure 1. Domain architecture and sequence analysis of full-length PCPBAe1.**
**(A)** Sequence architecture of full-length *Ae. aegypti* procarboxypeptidase B1 (*PCPBAe1*) showing the signal peptide (SP) (M*-Arg$^{1A}$), N-terminal pro-region (Arg$^{1A}$-Asp$^{75A}$), and a C-terminal mature peptidase domain (Asp$^7$-Phe$^{305}$) separated from the pro-region by an activation loop (Asp$^{75A}$-Asp$^7$). SP and pro-region cleavage sites are indicated with black arrows.
**(B)** Comparison of the sequences of regions in the activation segments of PCP that interact with the enzyme moiety. The sequences correspond, from top to bottom, to *PCPBAe1* (Ae1), procarboxypeptidase B from *H. zea* (*PCPBHz*) (Hz), the A1, A2, and B forms of procarboxypeptidases from bovine and porcine (Pb). The regions around residue 40 (left) and the C-terminus of the activation segment (right) are shown; the first residues of every sequence segment (34A and 66A) are indicated; "A" refers to residues belonging to the activation segment (pro-region). Strand *β*4 and helix *α*3 are present in all five sequences, whereas only the B form from bovine and porcine possesses a $3_{10}$ helix turn which is missing in *PCPBAe1* and CPBHz. Black and yellow arrows indicate the primary and secondary trypsin cleavage sites, respectively.

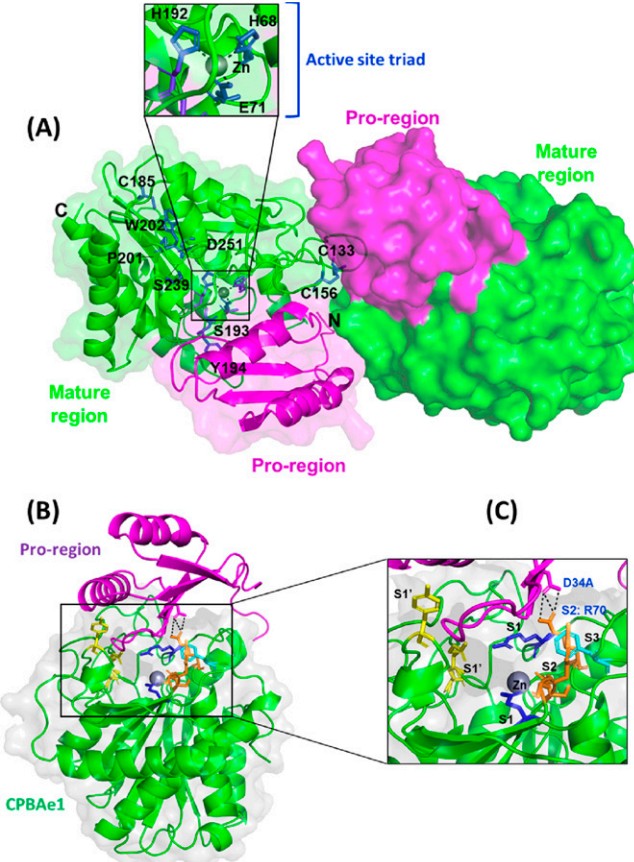

**Figure 2. Crystal structure of PCPBAe1 and the autoinhibition mechanism by the pro-region.**
**(A)** Overall structure of PCPBAe1 7EQX. There are two molecules in the asymmetric unit of the crystal. Pro-region (magenta) and mature peptidase domain (green) are indicated for each molecule in cartoon and surface representations. Close-up view of the active site pocket (inset) highlights the active site triad H[68], E[71] and H[192], coordinated by a zinc atom (grey spheres). A disulfide bond is shown at (C[133]-C[156]), whereas a free C[185] is also shown. Two cis-peptide bonds (S[193]-Y[194] and P[201]-W[202]) are shown with green. Important substrate specificity residues D[251] and S[239] are also indicated. **(B)** A single molecule of PCPBAe1 showing the auto-inhibition mechanism. **(C)** Close-up view of the substrate binding pocket of the mature enzyme with the subsites S₁' (Asp[251], Tyr[244], Asn[139], and Arg[140]) in yellow; S₁ (Glu[267] and Arg[123]) in blue; S₂ (Arg[70], Glu[94], Ser[193], Tyr[194], and Gly[195]) in light orange; and S₃ (Phe[276]) in dark orange are shown as sticks. Pro-region of PCPBAe1 (Asp[34A]) forms hydrogen bonds with the S₂ subsite (Arg[70]) (black dashed lines). Zinc atom is in grey.

region (Fig 2B). Residues Asp[34A] and Phe[35A] from the pro-region make contacts with the S₂ subsite residue Arg[70] (Fig 2C); this interaction is crucial for blocking small-peptide substrates from contacting the zinc atom, and thus maintain PCPBAe1 in its inactive form. In particular, the double hydrogen bond between Asp[34A] and Arg[70] might explain how PCPBAe1 remains completely inactive as compared to mammalian PCPA (22). Although other contacts are outside the region of substrate-binding residues, they are also close enough to block substrate binding. For example, Asn[30A] contacts Gln[164], which blocks the S₁ subsite residue Asp[168]. Similarly, Arg[47A] interacts with Gly[274], limiting access to the S₃ subsite Phe[276] (Fig 2C). Thus, by covering the active site and substrate-binding pocket, the pro-region regulates the activity of the enzyme and therefore, only the removal of the pro-region will allow access to substrate or inhibitor binding.

## Active site and substrate-binding pocket of the mature peptidase domain, CPBAe1

The mature CPBAe1 was generated by trypsin limited proteolysis of the full-length PCPBAe1, and further purified using size-exclusion and ion-exchange chromatography (Fig S1D). We tested the activities of the wild type (WT), single and double mutants of CPBAe1 (Asp[251]Glu, Ser[239]Gly, and Asp[251]Glu+Ser[239]Gly) using three carboxypeptidase substrates: Hippuryl-L-Arginine (Hip-L-Arg), Hip-L-Lys, and Hip-L-Phe (Table 1). The inactive enzyme PCPBAe1 was used as a negative control. CPBAe1 preferentially hydrolyzed Hip-L-Arg over Hip-L-Lys and displayed undetectable activity against Hip-L-Phe, which is a CPA-specific substrate (5, 23) (Table 1). The proenzyme PCPBAe1 showed no intrinsic activity against any of the substrates tested; this is contrary to the CPA family, which exhibit intrinsic activity against small-peptide substrates (24).

One zinc atom is present in each molecule of the structure and forms pentahedral coordination with the catalytic residues: His[68]N$^{\delta 1}$, His[192]N$^{\delta 1}$, Glu[71]O$^{\epsilon 1}$, and O$^{\epsilon 1}$ atoms and a water molecule. CPBAe1 contains one disulfide bond (Cys[133]–Cys[156]), which is the only conserved disulfide bond present in MCPs (10, 11, 20) (Figs 2A and S2). However, CPBAe1 has an additional free cysteine (Cys[185]) buried near the zinc-binding motif (Figs 2A and S2). Currently, there is no function ascribed to this cysteine residue. Moreover, CPBAe1 displays two of the three cis-peptide bonds (Ser[193]–Tyr[194] and Pro[201]–Trp[202]) present in mammalian MCPs (Figs 2A and S2) (10, 11, 20, 23, 25).

Combined with literature (10), we find that the key residues in the substrate-binding pocket of CPBAe1 include Asp[251], Ala[246], Try[244], Ser[239], Gly[203], Glu[170], Arg[140], and Arg[120] (Fig 2C). According to Titani et al, Tyr[244] and Arg[140] in CPBs anchor the terminal carboxyl group of substrates and prime it for hydrolysis (26). Other relevant residues for catalysis, such as Glu[170] and Arg[120], occupy similar positions in CPBAe1 compared with other MCP enzymes (Fig S2). Gly[203] of CPBAe1 is also conserved in both pancreatic and insect CPBs and CPAs, except for human pancreatic CPBh and porcine CPBp, where the residue in the equivalent position is occupied by Ser (Fig S2). The presence of Ser[239] in CPBAe1, which is conserved in *Helicoverpa zea* CPBHz and other mosquito CPBs, narrows the S₁' specificity pocket and affects the substrate-binding efficiency of the enzyme (Fig 3A and C). Of particular interest, Asp[251] at the bottom of the substrate specificity pocket in CPBAe1 (Fig 2A) is also critical for Arg-substrate hydrolysis (Table 1). Thus, our kinetics studies showed that the substrate specificity is dictated by amino acid residue combinations in positions 251 and 239 (Table 1).

So far, the only insect digestive CPB structure available is that of CPBHz from *H. zea* (11). CPBHz does not hydrolyze Arg-substrates, and has highly specific interaction towards Lys-substrates; this property was speculated to be dictated by the presence of Glu[255] (Asp[251] in CPBAe1) in place of Asp in CPBHz (11) (Fig S2). From our site-directed mutagenesis experiments, Asp[251]Glu mutation abrogated Arg-substrate hydrolysis completely in CPBAe1, whereas Lys-substrate hydrolysis remained unaffected (Table 1). Interestingly, the double mutant Asp[251]Glu and Ser[239]Gly restored Arg-substrate hydrolysis by CPBAe1, which was comparable to WT CPBAe1. Besides, Asp[251] and Ser[239]Gly further enhance Arg-substrate hydrolysis >30-fold as compared with WT CPBAe1 (Table 1). To further understand these substrate specificities, we compared Arg-hydrolyzing and non-Arg-hydrolyzing carboxypeptidases (Fig 3A). A detailed analysis

**Table 1. Substrate-hydrolysis of WT and mutant CPBAe1.**

| Enzymes/Substrates | 251 | 239 | Hip-L-Arg | Hip-L-Lys | Hip-L-Phe | PCPBAe1 |
|---|---|---|---|---|---|---|
| SEA/Units/mg | Asp | Ser | 17,000.34 | 11,600.41 | 0.00 | 0.00 |
| | Glu | Ser | 0.00 | 11,443.22 | 0.00 | 0.00 |
| | Asp | Gly | 25,012.10 | 11,216.91 | 0.00 | 0.00 |
| | Glu | Gly | 15,971.50 | 11,930.00 | 0.00 | 0.00 |

SEA, specific enzyme activity; 251 and 239 residues positions of WT and mutant CPBAe1.

of the substrate-binding pocket of CPBAe1 and CPBHz super-imposed onto the duck CPD-GEMSA (arginine analogue carboxy-peptidase inhibitor) complex (PDB: 1H8L) (27) revealed an apparent difference in the substrate-binding pockets of these enzymes (Fig 3A). Moreover, Ala[246], a key residue in the substrate-binding pocket of CPBAe1, is conserved among MCPs except in CPBHz, which has an Ile at this position (Fig S2).

### Comparison of PCPBAe1 structure with other procarboxypeptidase structures

A DALI structural homology search identified structural similarities among MCP proenzymes of the A/B subfamily of both insect and mammalian origins (Fig S3A–C and Table S1). Most of the secondary structures aligned well with the PCPBAe1 structure (Fig S3A–C). The nearest structural homolog of PCPBAe1 is the insect proCPA from *Helicoverpa armigera* (PCPAHa) (Fig S3A). PCPAHa aligns well with PCPBAe1 (PDB: 1JQG, 39% sequence identity, RMSD 1.4 Å for 394 Cα atoms) over the mammalian counterpart (PDB: 1KWM, 36% sequence identity, RMSD 1.6 Å for 394 Cα atoms) (Table S1).

Overall, most of the secondary structures of PCPBAe1 are well-conserved. However, an apparent rotation of ~8–10° is observed in the pro-region, perhaps because of the less-conserved nature of the residues in the pro-region (~20% sequence identity) compared with that of the mature peptidase domain (~40% sequence identity) of the homologs (PDBs: 1JQG, 1KWM, and 1NSA) (Figs S2 and S3A–C and Table S1).

### PCPBAe1 directly interacts with DENV-2 infectious virion, VLP, and E protein

Previous work suggested that PCPBAe1 inhibited DENV-2 release from the mosquito midgut cells, by binding to and hijacking the viral E protein/virion, thereby preventing the encapsulation of newly formed nucleocapsids or preventing virion maturation (6, 7). Interestingly, PCPBAe1 overexpression affects mature DENV-2 release from the mosquito C6/36 cell line, but not from mammalian Vero cells (7). Here, we conducted ELISA assays using recombinant PCPBAe1 purified from *Escherichia coli* (ECP) and insect baculoviral expression systems (ICP) (Fig S1A and C). ECP and ICP were

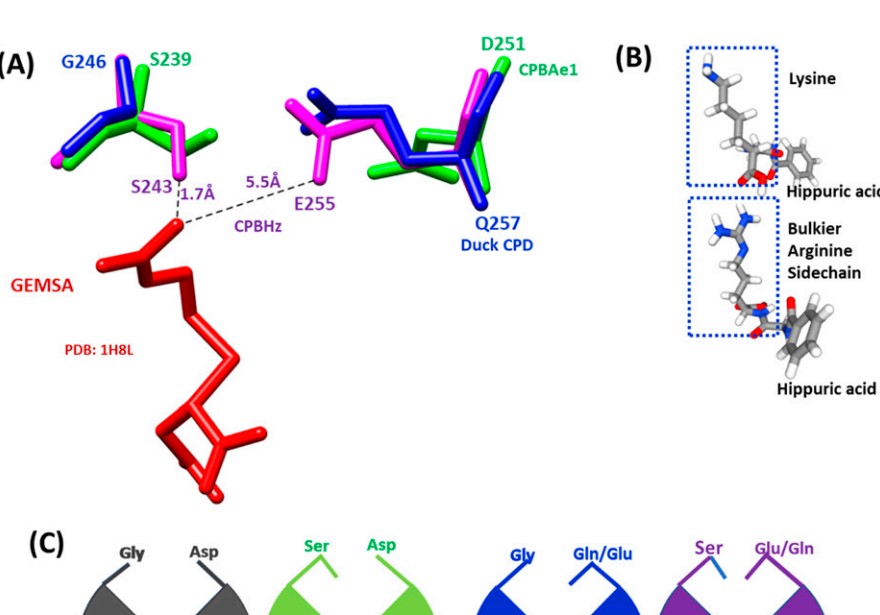

**Figure 3. Structural comparison of the substrate-binding pocket of CPBAe1 and CPBHz with duck CPD-GEMSA complex (Protein Data Bank [PDB]:1H8L).** **(A)** The substrate-binding pocket is governed by Asp[251] (for CPBAe1), Glu[255] (for CPBHz), and Gln[257] (for CPD) and Ser[239] (for CPBAe1), Ser[243](for CPBHz) and Gly[246](for CPD). The distance between GEMSA (red) and CPBHz are indicated. **(B)** Chemical structures of hippuryl-L-Arginine and hippuryl-L-Lysine showing the side chains. **(C)** Schematics showing the proposed substrate-binding architecture dictated by different combinations of amino acid residues.

incubated with DENV-2 E protein obtained from insect and mammalian expression systems (Fig S1F), VLPs purified from mammalian cells, and infectious virions of different DENV serotypes from the mosquito C6/36 cell line (Fig 4A). Different sources of proteins/VLPs/virions were used as a way to possibly uncover the mechanism behind the different effects of PCPBAe1 on DENV-2 release from different cell lines (7).

ECP and ICP interacted with insect- and mammalian cell-derived viral coat antigens/virions (Fig 4A). ICP showed marginally higher interaction with all the viral antigens/virions (~10–13-fold higher) as compared with ECP (Fig 4A). The reason for this higher interaction is unclear. Gel electrophoresis analysis showed that both ICP and ECP have similar molecular weights (Fig S1A and C). Hence, the difference in interaction is unlikely due to any post-translational modifications on ICP. We also noted that PCPBAe1 had a stronger interaction (~25–30-fold higher) with the insect-purified E protein than the mammalian-purified E protein/VLP (Fig 4A). This interaction disparity might be due to the source of the viral antigens, which likely have variations in post-translational modifications on the E glycoproteins. There are two $N$-glycosylation sites (Asn[67] and Asn[153]) on the DENV E protein (28), and the differences in the compositions of glycans between insect- and mammalian-purified glycoproteins (29) might play a role in the interaction disparities. The interaction with the peptidase domain (CPBAe1) or the pro-region (ProCP) alone (Fig 4B) (purified from *E. coli*; Fig S1D and E), led to a reduction in binding with DENV (>50-fold lower) suggesting that the full-length-PCPBAe1 is required for interaction. There was no interaction between PCPBAe1 and VLPs of chikungunya (CHIKV) and Mayaro (MAYV) viruses (two alphaviruses that are also transmitted by *Aedes* mosquitoes) (data not shown).

### Mapping the interaction regions of PCPBAe1 with the E protein using HDXMS

We used HDXMS and high-resolution structures to map the interacting regions of PCPBAe1 and the DENV-2 E protein (Fig 5A–C). We observed that the N-terminal pro-region of PCPBAe1 Ala[1A]–Asp[75A] (notably region Tyr[7A]–Ala[21A]) showed a single highest deuterium uptake than the peptidase domain Asp[7]–Phe[295] which exhibited multiple increased deuterium uptake (peptide clusters 171-182, 206-218 and 226-252) upon binding to E protein (Fig 5A and C). Only one locus (the linker between the pro-region and the peptidase domain [Pro[82A]–Leu[4]]) of PCPBAe1 showed significant protection against deuterium exchange in the PCPBAe1:E protein complex. Consequently, these identified sites were mutated to alanine or deleted from loop regions in each peptide and showed varying levels of reduction in the interaction as compared with wild-type PCPBAe1 (Fig 6). The extreme N-terminal exposed Asp[18A] and Glu[19A] deletion caused the most dramatic reduction in the interaction (~50-fold lower), followed by the charged residues mutations to alanine in the buried linker region (Pro[82A]-Leu[4]; Glu[85], Arg[87], and Arg[89]-AlaAlaAla) which showed moderate reduction in interaction (~20-fold) (Fig 6A). Deletion of the entire linker loop resulted in inclusion body formation and could not be purified and tested. The extreme C-terminal residues mutations Glu[168], Glu[170], Arg[172]-AlaAlaAla (peptide Gly[162]–Arg[172]), and Val[300]-Phe[305] deletion did not impact the interaction (Fig 6A).

PCPBAe1 primarily targets the exposed flexible loop regions of the E protein domains I/II; although, the interaction does not include the fusion loop (Asp[98]–Gly[109]) (Fig 5B and C). The peptide regions of E protein domains I/II (Lys[64]–Glu[84], Val[238]–Val[252], Leu[278]–Leu[287]) showed decreased deuterium exchange, suggesting that these regions are buried upon binding to PCPBAe1. Consequently, we used independent domains (domains I/II, and domain III; purified from *E. coli*) (Fig S1G and H) as truncated versions of the E protein to localize the importance of these regions in interacting with PCPBAe1. The interaction of the individual domains I/II or III alone with PCPBAe1 is significantly less (~70–85-fold lower) than that of the full-length E protein, suggesting that the full-length E protein is necessary for maximum binding (Fig 6B).

### PCPBAe1 binds to all DENV serotypes and 67NTT-67QTV and N153Q mutants

Sequence alignment of all four DENV serotype E proteins (Fig 7A) revealed some residue conservations among the buried peptides

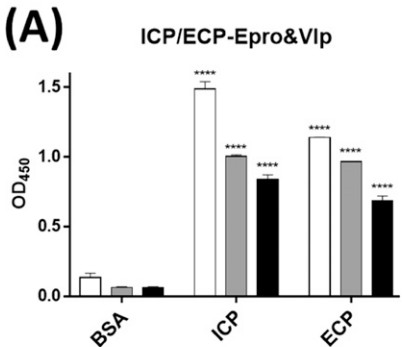

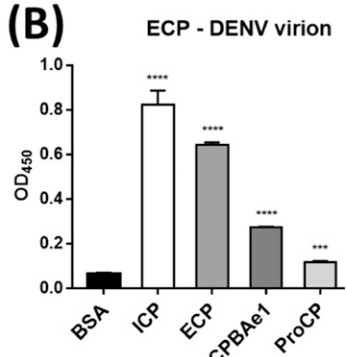

**Figure 4. PCPBAe1, CPBAe1, and ProCP interaction with DENV-2 E protein/virus-like particles or infectious virion.**
**(A, B)** All the interaction studies (panel A, B) were performed by ELISA in triplicates. **(A)** Insect cell-purified PCPBAe1 (ICP) and *E. coli* purified PCPBAe1 (ECP) interaction with insect cell purified E protein and mammalian HEK purified E protein/virus-like particles. **(B)** PCPBAe1 (ECP/ICP), mature peptidase domain (CPBAe1) and pro-region (ProCP) interaction with DENV-2 infectious virion. CPBAe1 and ProCP were produced from the *E. coli* expression system. Data were analyzed by independent unpaired t-test between two sets of data (control and treated samples), and by two-way ANOVA for multiple groups of data using GraphPad Prism version 7.00. Results are shown as mean ± S.D. *$P$ < 0.05, **$P$ < 0.01, ***$P$ < 0.001, and ****$P$ < 0.0001.

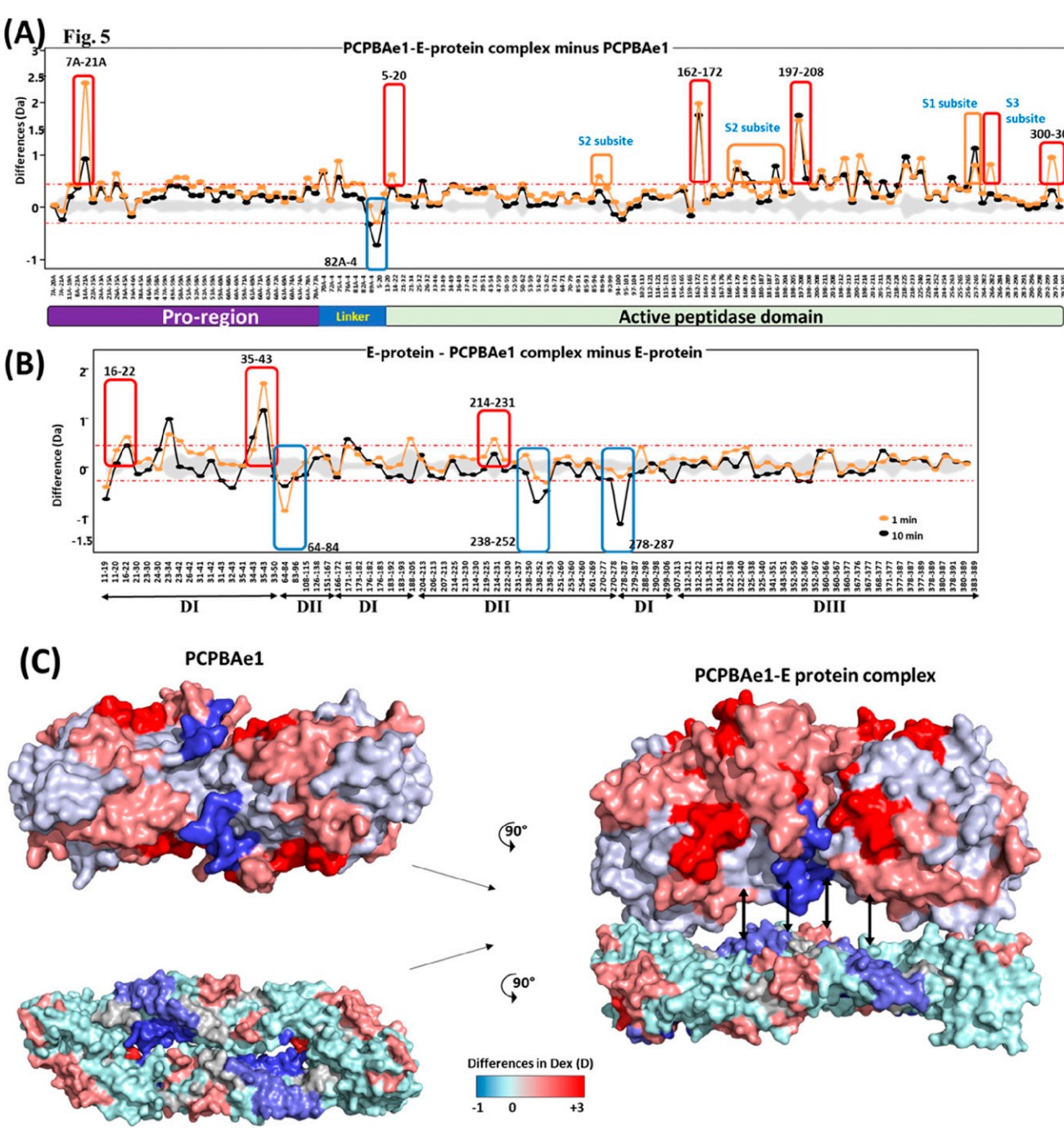

**Figure 5. Effect of E protein-PCPBAe1 binding on each protein and proposed binding mode.**
**(A)** Plot depicting the differences in deuterium exchange between E protein bound and free PCPBAe1 for various pepsin digested fragments at 1 min (orange) and 10 min (black) labeling timescales. Residue numbers spanning the N- to C termini of PCPBAe1 are indicated, as per the pro-region and peptidase domains. **(B)** Difference plot of differences in deuterium exchange (y-axis) between PCPBAe1-bound and free E protein for various pepsin digest fragments of E protein (x-axis). Residue numbers of various peptides with domain (I, II, and III) organization is shown. Peptide Ala$^{35}$-Asp$^{43}$ spanning domains I showed large-scale increased deuterium exchange in the complex. **(A, B)** Positive differences indicate increased deuterium exchange and negative values indicate decreased deuterium exchange in E protein bound to PCPBAe1 as compared with (A) PCPBAe1 alone or (B) E protein alone. Blue boxes highlight peptides of E protein or PCPBAe1 showing significant "protection" against deuterium uptake in the peptides in PCPBAe1:E protein complex. Each value is an average of three independent labeling measurements and the standard deviations are in grey. A threshold of ± 0.3 D is considered as significance cut-off and indicated by red-dashed lines. **(C)** Differences in deuterium exchange at 10 min labeling time are mapped on to the crystal structures of PCPBAe1 (PDB: 7EQX) (solved in this study) (left panel, top) and E protein (left panel, bottom) dimer (PDB: 1OAN), shown in surface representation. Difference maps are colored according to the key with regions showing decreased deuterium exchange in shades of blue and increased deuterium exchange in shades of red. A surface representation model of PCPBAe1-E protein complex (right panel) showing the interaction interface is shown, with the predicted regions of interaction indicated by double-headed arrows.

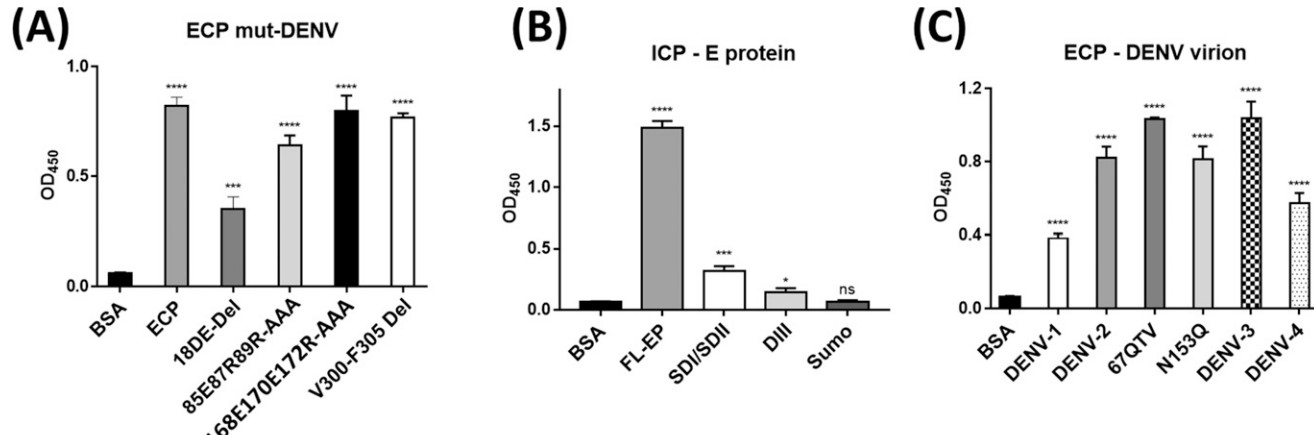

**Figure 6. WT PCPBAe1, mutant PCPBAe1, DENV-E protein, and infectious virion interaction.**
**(A, B, C)** All the interaction studies (panels A, B, C) were performed by ELISA in triplicates. **(A)** Wild-type and mutants of ECP interaction with DENV-2 infectious virion. **(B)** Insect-purified full-length E protein and *E. coli* purified E protein domains (DI/II) and (DIII) interaction with ICP. **(C)** DENV serotypes and mutants interaction with ECP. Data were analyzed by independent unpaired t-test between two sets of data (control and treated samples), and by two-way ANOVA for multiple groups of data using GraphPad Prism version 7.00. Results are shown as mean ± S.D. *$P < 0.05$, **$P < 0.01$, ***$P < 0.001$, and ****$P < 0.0001$.

(Lys$^{64}$–Glu$^{84}$, Val$^{238}$–Val$^{252}$, and Leu$^{278}$–Leu$^{287}$) and among the peptides that exhibited increased deuterium exchange (Ser$^{16}$–Asp$^{22}$, Ala$^{35-43}$, and Leu$^{214}$–Trp$^{231}$) (Fig 5B). ELISA experiments with the four DENV serotypes showed different levels of interaction in the following decreasing order: DENV-3 > DENV-2 > DENV-4 > DENV-1 (Fig 6C). From the sequence alignment, differences in key amino acid residues of the E protein interacting regions (Fig 7B) may be responsible for the observed differences in affinity (Fig 6C). A protein-wide overview of the HDXMS data indicated that many regions showed higher deuterium exchange near the E protein glycosylation site Asn$^{67}$ and not Asn$^{153}$. To test the effect of Asn$^{67}$ and Asn$^{153}$ in the interaction between DENV-2 and PCPBAe1, we generated partially deglycosylated DENV-2 mutants that lack their glycan motif either at N67 (NTT-QTV) or at N153 (N153Q). We then conducted an ELISA assay with 67NTT-67QTV and N153Q mutant DENV-2 virions. From Fig 6C, the 67QTV mutant displayed statistically insignificant stronger interaction with PCPBAe1 than the WT, whereas the N153Q mutant interaction was similar to the WT.

## Discussion

MCPs have adopted a complex function in mosquitoes, with roles in mosquito reproduction (5), enhancing malaria parasite infection (4), but also suppressing DENV replication in the mosquito midgut (7). Here, we determined the first crystal structure of a mosquito MCP, PCPBAe1, and characterized the substrate hydrolysis properties of the mature peptidase domain (CPBAe1). We identified critical differences between CPBAe1 substrate hydrolysis and that of mammalian and other insects MCPs. Furthermore, through a combination of HDXMS, ELISA, and mutational studies, we described and proposed how PCPBAe1 binds to and suppresses DENV infection.

CPBAe1 hydrolyzes Arg- and Lys-substrates, and this substrate specificity is dictated by residues Asp$^{251}$ and Ser$^{239}$. From the comparison of substrate-binding pocket, the presence of Gly$^{239}$ compensates for Glu/Gln$^{251}$, whereas Asp$^{251}$ compensates for Ser$^{239}$ to accommodate Arg-substrate hydrolysis. Although Asp and Glu have similar physicochemical properties, their side chains affect Arg- and Lys-substrates binding and hydrolysis differently. The Arg side chain of Arg-substrates is bulkier and does not insert easily into the substrate-binding pocket compared with the less bulky Lys side chain. Thus, Glu$^{251}$/Ser$^{239}$ severely restricts the substrate-binding pocket and does not allow accommodation of the bulky guanidyl group of peptides containing C-terminal Arg. On the contrary, Asp$^{251}$/Gly$^{239}$ allows the side chain of a C-terminal Arg residue of a bound substrate to move away from Asp$^{251}$, which increases Arg-substrate hydrolysis.

Sequence alignment shows that Asp$^{251}$ and Ser$^{239}$ are conserved across mosquito CPBs, and these amino acid residues might be critical for blood digestion and in facilitating egg development (11, 19). This might explain the poor fecundity observed in female mosquitoes when CPB activity was blocked (11, 19). Thus, blocking the enzymatic activity of CPBs in mosquitoes could contribute as a mosquito population control strategy.

Solving the structure of PCPBAe1 allowed us to investigate how the full-length PCPBAe1 binds to the DENV-2 E protein to suppress infection. In the absence of PCPBAe1, ingested DENV undergoes normal replication in the midgut (Fig 8A). During blood ingestion, PCPBAe1 becomes activated (CPBAe1) to digest blood proteins. It is possible that the mature CPBAe1, when released into DENV-infected blood in the midgut lumen, binds to mature virions and partially inhibits binding and entry into midgut cells. We surmise that both the full-length-PCPBAe1 and the mature form (CPBAe1) could potentially bind to the virus inside the mosquito (Fig 8B); albeit the full-length-PCPBAe1 might exhibit a stronger inhibition potency against virus propagation due to its ability to bind more strongly (~50-fold stronger) to the virus. The weaker interaction of the mature peptidase domain alone in our in vitro assay might be due to the disruption of the putative interaction linker region (Tyr$^{4A}$–Glu$^{21A}$) to produce the active enzyme. The N-terminal peptide region

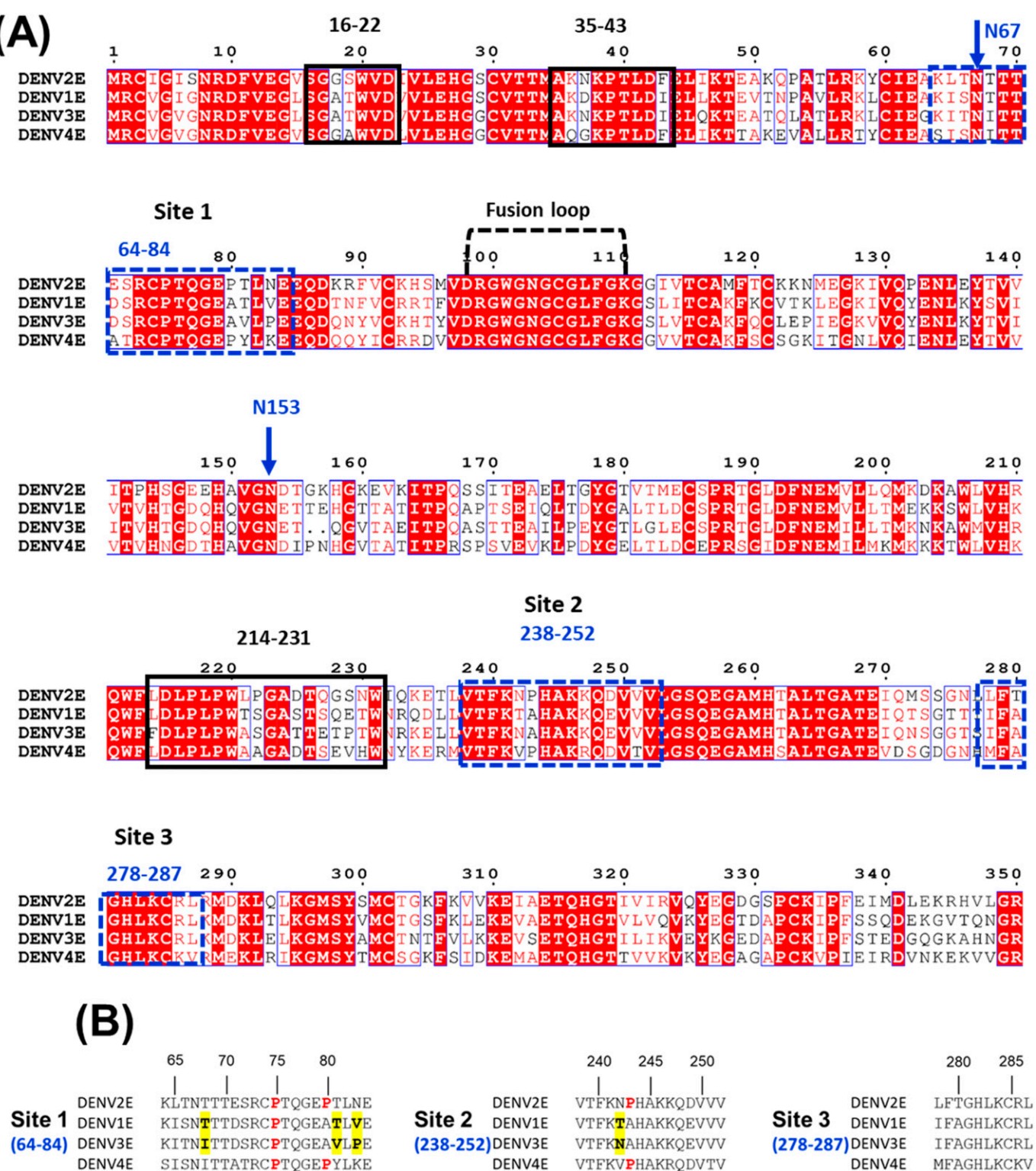

**Figure 7. Sequence alignment of E protein from the four DENV serotypes.**
**(A)** The E protein peptides that are buried upon binding to PCPBAe1 are indicated in blue dash boxes spanning protein domains I/II (a.a 64–84, 238–252, and 278–287), whereas peptide regions that underwent increased deuterium exchange are indicated in black boxes spanning E protein domains I/II (a.a 16–22, 35–43, and 214–231). The Fusion loop is also indicated. The two N-glycosylation sites (N67 and N153) are shown with blue arrows. White fonts with red background show conserved residues, whereas red fonts with white background are areas of high similarities. The alignment was done with Clustal Omega and ESPript 3.0. **(B)** Zoomed-in sequence alignment of E protein regions targeted by PCPBAe1. Important amino acid residue changes are indicated with bold font and yellow highlights. Conserved proline residues are indicated with bold red font.

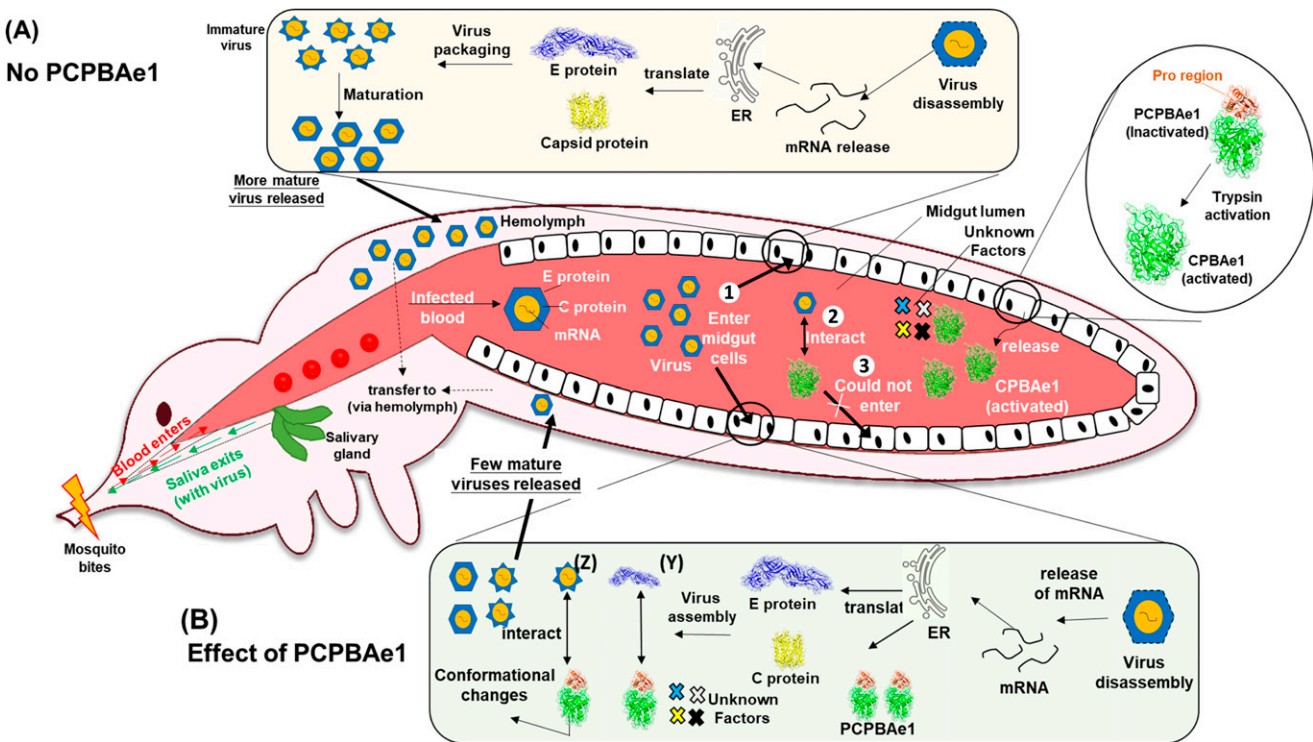

**Figure 8. A schematic of the binding and inhibition mechanism of PCPBAe1/CPBAe1 on DENV-2 in the mosquito midgut.**
Female mosquito takes in an infected bloodmeal containing DENV. **(1)** Some DENV particles enters midgut **(2)** Bloodmeal in the midgut causes overexpression of procarboxypeptidase (PCPBAe1) and activated by trypsin into CPBAe1 (inset, circle), which binds to mature DENV and **(3)** partially inhibits cellular binding and entry. **(A)** DENV proliferate as immature viruses, which undergo conformational changes to become mature viruses in PCPBAe1-free cells. Mature viruses are eventually released into the salivary gland and subsequently into the mosquito saliva. **(B)** In the midgut epithelia, overexpressed PCPBAe1 binds to: **(Y)** E protein and prevents viral nucleocapsid packing, **(Z)** immature viruses and prevents conformational changes required for virus maturation. These events lead to a low number of released mature viruses into the hemolymph. Unknown factors possibly exclusive to the mosquito cellular milieu that may augment PCPBAe1 inhibitory activity against DENV are shown as different colors of X in both the midgut cell and the midgut lumen.

(Tyr[4A]–Glu[21A]) of PCPBAe1 seems to be targeted primarily by the E protein, as shown by the significant deuterium exchange at this site, and further corroborated by the alanine site-directed mutagenesis of residues Asp[18A] and Glu[19A]. However, using the pro-region alone (ProCP) shows an extreme reduction in interaction (~60-fold reduction), suggesting that the conformational dynamics required for interaction of full-length PCPBAe1 might be different from the individual pro-region. The identified DENV E protein-interacting peptide regions of PCPBAe1 in other *Ae. aegypti* PCPs (PCPAe1 and PCPBAe111) are poorly conserved, suggesting that PCP homologs may possess different levels of inhibitory activities, if any, against DENV.

Furthermore, the HDXMS results suggest that PCPBAe1 binds across the E protein's dimeric interface by targeting flexible loop regions but not the conserved fusion loop (Asp[98]–Lys[110]). The 4G2 anti-DENV E protein antibody used in our ELISA experiment targets the conserved fusion loop epitope, consistent with the fact that the antibody detects DENV E-protein/virion/VLP even in the presence of PCPBAe1. In the previous docking studies by Tham et al, the authors predicted some of the residues of the E protein (Thr[66], Asn[67], Val[251], and Lys[122]) that might interact with PCPBAe1 ([30], [31]). Our HDXMS analysis mapped all of these residues except Lys[122] ([30], [31]), and additional E protein amino acid stretches from domains I/II (Lys[64]–Glu[84], Val[238]–Val[252], and Leu[278]–Leu[287]). Domain III, however,

did not appear to contribute to the interaction with PCPBAe1. Thus, the individual domains alone are insufficient for interaction, as the full-length E protein shows a more substantial interaction. Targeting the individual E protein subunits or the mature/immature virion particle could lead to: (1) unavailability of E protein required for encapsulating newly formed viral nucleocapsids in the midgut, (2) prevent the E protein rearrangement and impede the maturation of viral particle or restrict the overall budding/exiting of viruses from the midgut cells, or (3) impede cellular binding/endocytosis by binding of the mature CPBAe1 to the virus in the midgut lumen.

The interaction between PCPBAe1 and all DENV serotypes suggests that PCPBAe1 could potentially suppress all serotypes in the mosquito. There are three discontinuous interaction sites on the DENV E protein that bind to PCPBAe1. We systematically compared the sequences of these sites to understand the differential interactions (DENV-3 > DENV-2 > DENV-4 > DENV-1) with DENV serotypes. At first, we compared the sites of DENV-3 and DENV-1, which are at the opposite ends of affinities. Site-3 is completely conserved between DENV-3 and DENV-1, whereas site-2 shows a single residue change (Thr[242]Asn) (Fig 7B). In contrast, site-1 shows several key differences. Specifically, the two hydrophilic Thr residues to hydrophobic residues in DENV-3 (Thr[68]Ile and Thr[81]Val). These hydrophobic residues may enhance the interaction due to the

tendency to bury themselves in the interaction surface. In addition, a Pro[83] residue replaces Val[83] in DENV-3, creating a proline bracket with Pro[78] residue (32). Such proline brackets are known to enhance protein–protein interactions (33). Thus, it appears that the Site-1 (62-82 residues) mostly defines the affinity differences observed between DENV-3 and DENV-1. There are several amino acid residue differences observed in all three sites between DENV-2 and DENV-4. Interestingly, sites-1 of both DENV-2 and DENV-4 have slightly shorter regions within the proline brackets than DENV-3. They also have a conserved Pro[243] residue unlike DENV-1 and DENV-3. It is unclear which of these changes are responsible for the observed differences in affinity in these two serotypes.

Only the glycan motif at Asn[67] on the E protein seems to influence interaction. The different glycan compositions at Asn[67] between the insect and mammalian cell-purified viral antigens might partially account for the marginal differences observed in the interaction between PCPBAe1 and DENV (29). However, because this difference is only marginal (and not significant), these findings cannot wholly explain why PCPBAe1 only suppresses the release of DENV-2 in mosquito C6/36 cell line but not in a mammalian Vero cell line (7). Thus, apart from PCPBAe1 interaction with the E protein, other conditions in the mosquito cell line microenvironment such as cellular proteins/receptors or attachment factors (34), that are different from the mammalian cell line microenvironment likely play a role in virus inhibition (Fig 8). For example, virus internalization might occur through distinct entry pathways, including clathrin-mediated or non-classical clathrin-independent endocytosis, depending on whether mosquito or human cell or specific host cells (34). Furthermore, C6/36 is a better cell line for DENV infection than Vero cell line (34).

In summary, we show that the residues Asp[251] and Ser[239] control substrate-specificity, and such residues may be targeted for developing anti-CPB molecules to arrest digestion and control mosquito reproduction. Because of its ability to bind all DENV serotypes, PCPBAe1 may also inhibit other DENV serotypes by targeting the E protein/virion, which could hamper viral packaging or morphogenesis, leading to an overall reduced viral maturation and release from the mosquito midgut. Alternatively, activated CPBAe1 may also inhibit mature DENV when released into DENV-infected blood in the midgut lumen, although this inhibition may be less than that of the full-length PCPBAe1. Besides, because the mature region usually degrades the pro-region of procarboxypeptidases upon tryptic cleavage, the independent pro-region may not be available in the midgut when cleaved from the proenzyme (35). However, PCPBAe1 might be specific to DENV as we did not observe PCPBAe1 binding to CHIKV or MAYV VLPs.

Thus, our structural and functional studies of PCPBAe1 provide a blueprint for developing two distinct strategies: (i) carboxypeptidase inhibitors or antibodies as a strategy for mosquito population control and (ii) how PCPBAe1 elicits antiviral properties against DENV. However, it would be essential to evaluate the strategies aimed at blocking carboxypeptidase activities in DENV endemic areas to avoid compromising CPB antiviral activities in the mosquito. Moreover, even though PCPBAe1 serves as an anti-dengue viral molecule in the mosquito, it may not represent an effective dengue or other mosquito-borne viral diseases control strategy as the inhibition of viral propagation seems to only occur in the vector and may be specific to DENV.

# Materials and Methods

## Cloning, recombinant protein expression and purification

### PCPBAe1 and E protein domain I/II and III expression and purification from the E. coli system

The gene coding for proenzyme *Aedes aegypti* procarboxypeptidase B1 (*PCPBAe1*) (Genbank accession number: Q6J661) was codon optimized and synthesized by Genscript for recombinant protein expression in *E. coli* (BL-21(DE3)) (shuffle) cells. The gene was subcloned into pGEX-6p-1 vector (Novagen) using the Bamh1/Xho1 restriction sites to form the pGEX-precision protease site-*PCPBAe1* fusion construct. Confirmed fusion plasmid was transformed into the *E. coli* shuffle (BL-21(DE3)) cells. Transformed cells were inoculated into 10 ml LB medium supplemented with 100 $\mu$g/ml ampicillin and grown at 37°C overnight as a primary culture inoculum. The overnight culture was used to inoculate 1L flask secondary LB medium and grown until the optical density (OD) reached 0.8. The culture was cooled at room temperature and induced with 0.3 mM IPTG, supplemented with 60 $\mu$M zinc sulphate. The induced culture was shaken at 18°C, 150 rpm overnight. Overnight bacteria culture was pelleted down at 4,000 rpm (centrifuge rotor ID: JLA-8.1000) for 30 min. Bacteria cell pellets were resuspended in 35 ml/l pellet in lysis buffer containing 50 mM citrate buffer, pH 6.0, 0.3 M NaCl, 5% glycerol 1 mM DTT, and a protease cocktail inhibitor tablet (Roche). Pellets were vortexed and sonicated for three cycles (5 min, 1 s on, 2 s off). Sample was then span down at 18,000 rpm (centrifuge rotor ID: JA-20) for 30 min at 4°C and bound to GST beads (GE) previously equilibrated with the same lysis buffer at 4°C for 4 h for affinity purification. Beads were washed three times with 90 ml of lysis buffer each time and incubated overnight with elution buffer supplemented with PreScission Protease (GE) to cleave the GST tag. First step purification of anion exchange chromatography was done on QHP column (GE), applying a gradient from 100% buffer A (20 Mm Tris, pH 8.0, 1 mM DTT) to 100% buffer B (20 mM Tris/NaCl 1 M, pH 8.0, 1 mM DTT) in 30 min. A final size exclusion chromatography on a Superdex S-200 column (GE) (20 mM sodium citrate pH 6.0, 0.1 M NaCl, 5% glycerol, and 1 mM DTT) was performed to achieve maximal purity. The peak fractions were collected and concentrated to 10 mg/ml. All the other PCPBAe1 mutants were purified using the *E. coli* expression and purified as above.

DENV-2 E protein domains I/II (a.a 1–296) and domain III (a.a 296–394) were cloned separately into a modified pet23b vector carrying sumo-tag and modified pet32a vector carrying only 6X-Histidine tag. Protein expression conditions were similar to the above described. Purification was carried out with nickel beads affinity chromatography, followed by ion exchange chromatography (buffer similar to above described) and gel filtration in the buffer condition (Tris, pH 8.0, 150 mM NaCl, 5% glycerol, and 1 Mm DTT). The tags were not cleaved before further assays.

### PCPBAe1 expression and purification from the insect baculoviral system

The *PCPBAe1* construct was further cloned into pFastBac1 vector with 6X-Histidine purification tag for protein expression in insect-cell baculovirus expression system using *Spodoptera frugiperda*

(SF9) cells by following previous protocols (36, 37). Protein was secreted into the culture medium and purified using nickel-NTA beads and gel filtration chromatography. Dynamic light scattering studies were carried out on a DynaPro Light Scattering instrument (Protein Solutions) to assess protein quality.

### Enzymatic assay of CPBAe1

The proenzyme PCPBAe1 was activated using trypsin limited proteolysis into the active form CPBAe1 and further purified using gel filtration and ion exchange chromatography. CPBAe1 activity was determined by the continuous spectrophotometric rate determination method of Folk et al (1960) where the reaction velocity is determined by an increase in absorbance at 254 nm resulting from the hydrolysis of Hippuryl-L-Arginine (Hip-L-Arg), Hip-L-Lys, and Hip-L-Phe. One unit causes the hydrolysis of 1 $\mu$mol of Hippuryl-L-arginine per minute at 25°C and pH 7.65 under the specified conditions. Briefly, substrates at different concentrations (0.05–10 mM) were diluted in buffer (0.025 M Tris–HCl, pH 7.65, containing 0.1 M sodium chloride). Different concentrations of stock trypsin activated CPBAe1 solution were diluted in assay buffer (0.025 M Tris–HCl buffer, pH 7.65, containing 0.1 M NaCl). The spectrophotometer was set at 254 nm and 25°C. Next, 2.9 ml of substrate solution was pipetted into two separate quartz cuvettes and incubated in spectrophotometer at 25°C for 3–4 min to reach temperature equilibration and blank rate was established. To start activity measurement, 0.1 ml of diluted enzyme was added to the substrate in the cuvette and the increase in absorbance (A254) was recorded for 3–4 min. The fastest linear rate ($\Delta$A254nm/minute) over a 1-min interval for the test and the blank reactions was then determined from the initial linear portion of the curve and used to calculate the specific enzyme activity values. All the OD readings were done in triplicates. The $K$m was also calculated using a kinetics plot.

### Crystallization and structure determination of proenzyme PCPBAe1

Crystallization screens were conducted using the sitting drop vapor diffusion method at room temperature (22°C) by mixing protein with the crystallization reservoir solutions at a 1:1 volume ratio. Crystals appeared after about 2 wk in the condition Crystal Screen 1(0.2 M ammonium sulphate, 0.1 M sodium cacodylate trihydrate, pH 6.5, 30% wt/vol PEG 8000) (Hampton Research). The diffraction quality crystals were immersed in mother liquor solution supplemented with 25% ethylene glycol as a cryoprotectant for data collection. A complete native 2.0 Å resolution diffraction data set was collected at 100K using Rigaku MicroMax-007 HF equipped with Saturn 944+ CCD detector, National University of Singapore. Data were processed with the HKL-2000 program (38). The structure was determined using molecular replacement (39) using the coordinates of CPAHa (PDB: 1JQG) (10) and the final model was refined at 2.08 Å resolution. The structure has good stereochemical parameters evaluated with PROCHECK (Table S2). PyMOL and CHIMERA (40) were used to prepare all structure-related figures.

### ELISA assay to characterize PCPBAe1-DENV virion, E protein and VLP interaction

High binding 96-well plates (Thermo Fisher Scientific) were coated overnight at 4°C with a 5 $\mu$g of all PCPBAe1 constructs and mutants or 5 $\mu$g BSA or sumo tag protein alone in coating buffer. All samples were coated in triplicates. The next day, plates were washed twice with PBS and blocked with blocking buffer (1% BSA in 1XPBST) for 1 h at 28°C. Plates were incubated with 100 $\mu$l of DENV-2 VLPs/infectious virion serotypes (DENV-1, DENV-2, DENV-3, and DENV-4) (2 × 10$^5$ p.f.u) or recombinant DENV-2 E protein (1 $\mu$g) (full-length, sumo-tagged domain I/II, or domain III) at 28°C for 1 h, considering the temperature of the mosquito vector. DENV-2 E protein (DENV-2-ENV-100) and VLPs (DENV-2-VLP-100, CHIKV-VLP-100, and MAYV-VLP: REC31616-100) were purchased from The Native Antigens Company (purified from HEK293 cell cultures), whereas infectious virion was obtained from our collaborators (A/P Sylvie Alonso, National University of Singapore). Unbound proteins/virion were washed three times with buffer (1× PBST+ 0.01% Tween 20) and incubated with 100 $\mu$l of 1 $\mu$g/ml mouse anti-DENV-2 primary antibody (MAB10216, Millipore or anti-E protein domain III [2D73], absolute antibody) for 2 h at RT to detect VLP/E protein/virion. Unbound primary antibody was then washed three times with buffer (1× PBST+ 0.01% Tween 20) and plates incubated with 100 $\mu$l of 1:5,000 dilution rabbit anti-mouse HRP conjugated secondary antibody and washed six times after 2 h of incubation at RT. Reaction was visualized after incubating with 80 $\mu$l of tetramethylbenzidine (TMB). After 5 min, the reaction was stopped with 1 M HCl and plates were read at 450 nm in Tecan Infinite¬ 200 pro (Life Sciences and Diagnostics).

### Hydrogen-deuterium exchange (HDXMS) of PCPBAe1-E protein interaction

Interaction of PCPBAe1 and DENV-2 E protein were mapped by amide hydrogen-deuterium exchange mass spectrometry (HDXMS) (41). HDX reaction was initiated by diluting the protein in 90 $\mu$l deuterated buffer (50 mM PBS pH 7.4) to a final 90% D$_2$O (Cambridge Isotopes) concentration. For HDXMS of free proteins, 3 $\mu$l of 1 mg/ml of DENV-2 E protein and 5 $\mu$l of 1 mg/ml of PCPBAe1 were used. For HDXMS of complex, the proteins were mixed in 1:2 E protein: PCPBAe1 stoichiometric ratio and incubated for 20 min before deuterium labeling reaction. Deuterium labeling was carried out for 1- and 10-min time points at 28°C temperature, similar to mosquito vector. The exchange reaction was stopped by lowering the pH to ~2.6 by addition of 100 $\mu$l quench solution (0.8 M guanidinium hydrochloride, 0.25 M TCEP) and temperature to 0°C to minimize back-exchange. Non-deuterated control experiments of PCPBAe1 and E protein alone were also carried out by diluting the samples in aqueous buffer, followed by quench solution.

The quenched samples were incubated on ice (0°C) for 30 s followed by 3 min proteolytic digestion by immobilized pepsin and the digested pepsin-cleaved fragments ("peptides") were resolved by reverse-phase chromatography using C18 column. The peptides were eluted using a 10 min 8–40% gradient of 0.1% formic acid in acetonitrile, pumped at 40 $\mu$l/min by nanoACQUITY binary solvent manager (42). The peptides eluted were then subjected to mass

analysis by injecting onto a coupled high resolution Synapt G2-Si mass spectrometer and identified in HDMSe mode, as described previously (41).

Mass spectra of non-deuterated controls were used for peak identification and peptide matching using ProteinLynx Global Server v3.0.1 software (Waters) (43) against individual databases consisting amino acid sequences of PCPBAe1 and DENV-2 E protein. Peptides were filtered and only high signal-to-noise ratio non-overlapping peptides were selected for analysis of deuteration by DynamX v3.0 (Waters). All values reported are an average of three deuterium exchange reactions and are not corrected for back-exchange. The absolute difference in deuterons exchanged by all peptides relative to another condition is represented as a difference plot listed from the N- to C terminus.

### Site-directed mutagenesis

For the activity assays, three mutants were generated: Asp$^{251}$Glu, Ser$^{239}$Gly, and Asp$^{251}$Gly+Ser$^{239}$Gly. Based on our HDXMS results, four *PCPBAe1* mutants were generated including Asp$^{18A}$ and Glu$^{19A}$ deletion, linker region (Glu$^{85}$Ala Arg$^{87}$AlaArg$^{89}$Ala) and (Glu$^{168}$Ala, Glu$^{170}$Ala, and Arg$^{172}$Ala) alanine mutagenesis, and Val$^{300}$-Phe$^{305}$ deletion. All the primer/oligonucleotide information are provided in Table S3. All the PCR reactions were carried out using KAPA-BIOSYSTEMS HiFi PCR reaction Kits (Roche).

### Generation of de-glycosylated DENV-2 mutant

D2Y98P-PP1 (GenBank: JF327392.1) RNA genome was extracted using QIAamp Viral RNA Kits (QIAGEN). cDNA synthesis was performed using GoScript Reverse Transcriptase (Promega) as per manufacturer's instructions. Four PCR fragments of around 2,700 nucleotides long were generated from cDNA using primer pairs with NEB Q5 Hot-Start high-fidelity 2× Master Mix (New England Biolabs). Fragments were gel-purified with MinElute gel extraction Kit (QIAGEN) after agarose gel electrophoresis and blunt end cloning was performed into pCR-Blunt II-TOPO Vector (Invitrogen). Site-directed mutagenesis was performed on plasmid that carried E gene to introduce the desired mutation using primer pairs. The mutated gene was amplified with other genes fragments via PCR and seamlessly assembled with vector contains CMV promoter sequence, hepatitis delta virus ribozyme, and simian virus 40 (SV40) poly-A sequence using NEBuilder HiFi DNA Assembly Master Mix (New England Biolabs) at 50°C for 60 min. This CMV-DENV genome-HDVr-SV40pA assembled product was transfected into BHK-21 cells using Lipofectamine 2000 (Invitrogen). Four to 6 d later, the viral supernatant was collected and sequenced for validation using Sanger sequencing. All the primer/oligonucleotide information are provided in Table S4.

## Data Deposition

Three dimensional atomic coordinates, and structure factors of PCPBAe1 have been deposited in the Protein Data Bank, www.pdb.org (PDB ID codes 7EQX).

## Supplementary Information

## Acknowledgements

This work was supported by Ministry of Education, Singapore (MoE Tier-3) grant (R154-000-697-112) and R154-000-C07-114 (AcRF Tier 1 grant), respectively. E Gavor is a graduate scholar in receipt of the Singapore International Graduate Award (SINGA) research scholarship.

### Author Contributions

E Gavor: conceptualization, data curation, software, formal analysis, validation, visualization, methodology, and writing—original draft, review, and editing.
YK Choong: data curation, software, formal analysis, validation, visualization, methodology, and writing—review and editing.
NK Tulsian: data curation, software, formal analysis, validation, visualization, methodology, and writing—review and editing.
D Nayak: software, formal analysis, validation, and visualization.
F Idris: formal analysis, validation, investigation, methodology, and writing—review and editing.
H Sivaraman: methodology.
DHR Ting: methodology.
A Sylvie: resources, formal analysis, and writing—review and editing.
YK Mok: writing—review and editing.
RM Kini: formal analysis, validation, visualization, and writing—review and editing.
J Sivaraman: conceptualization, resources, software, formal analysis, supervision, funding acquisition, validation, project administration, and writing—original draft, review, and editing.

### Conflict of Interest Statement

The authors declare that they have no conflict of interest.

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
