## [Reviewer comments · Life Science Alliance]

Life Science Alliance

Structure of *Ae. aegypti* procarboxypeptidase B1 and its binding with DENV for controlling infection

Edem Gavor, Yeu Choong, Tulsian Nikhil K, Digant Nayak, Fakhriedzwan Idris, Hariharan Sivaraman, Donald Ting, Alonso Sylvie, Yu-Keung Mok, R. Manjunatha Kini, and J. Sivaraman

DOI: <https://doi.org/10.26508/lsa.202101211>

Corresponding author(s): J. Sivaraman, National University of Singapore

Review Timeline:

Submission Date:	2021-08-24
Editorial Decision:	2021-09-23
Revision Received:	2021-10-06
Editorial Decision:	2021-10-07
Revision Received:	2021-10-18
Accepted:	2021-10-19

Scientific Editor: Novella Guidi

Transaction Report:

September 23, 2021

Re: Life Science Alliance manuscript #LSA-2021-01211

Prof. J Sivaraman
National University of Singapore
Department of Biological Science
14 Science Drive 4
Singapore 117543

Dear Dr. Sivaraman,

Thank you for submitting your manuscript entitled "Structural basis for the impact of *Aedes aegypti* procarboxypeptidase B1 on Dengue virus replication" to Life Science Alliance. The manuscript was assessed by expert reviewers, whose comments are appended to this letter. We invite you to submit a revised manuscript addressing the Reviewer comments.

Thank you for this interesting contribution to Life Science Alliance. We are looking forward to receiving your revised manuscript.

Sincerely,

B. MANUSCRIPT ORGANIZATION AND FORMATTING:

Reviewer #1 (Comments to the Authors (Required)):

The essential roles of midgut metalloproteases (MCPs) have been previously established and PCPBAe1 has been shown to regulate dengue virus accumulation and release from midgut cells. Due to these crucial roles the Gavor et al have undertaken a structural and molecular characterisation of the full-length PCPBAe1 (proenzyme) and the mature peptidase domain (CPBAe1) and interactions with the dengue envelope (E) protein. Gavor et al present the crystal structure of the PCPBAe1 to 2.08 Å resolution (PDB: 7EQX). The validation report has not been provided however a brief look over the data collection and refinement statistics shows that completeness is low in the high-resolution shell and perhaps the data should be cut at a slightly lower resolution (2.1 or 2.2 Å). An analysis of the interactions mediating pro binding is provided and specific interactions occluding access to the substrate binding region are noted. Substrate binding was explored by structural comparison and site directed mutagenesis to demonstrate the mechanism by which CPBAe1 accommodates Arg substrates.

Previous yeast-2-hybrid studies have established an interaction occurs between PCPBAe1 the dengue E protein/virion resulting in intracellular virus particle accumulation. Binding was measured by ELISA for combinations of insect (ICP) and E coli (ECP) produced PCPBAe1 and DENV2 E protein. ICP showed a higher interaction with all the viral antigens/virions as compared with ECP. Whilst the ICP and ECP proteins show similar molecular weights in the SDS-PAGE there is a lower molecular weight band in the ICP (Supp Fig 1C) sample, this sample should be investigated by mass-spectrometry to determine whether this is a contaminant or the mature CPBAe1 protein.

In the discussion the authors explain that the difference in glycan composition at Asn67 might explain the differences in suppression of DENV2 release from insect C6/36 and mammalian Vero cells. This is insignificant and as Gavor et al note "other conditions present in the mosquito cell line microenvironment....are likely involved in virus inhibition". Tham et al (2014) also reported an increase in extracellular DENV2 within the Vero cell line when treated with PCPBAe1 and this would preclude the use of PCPBAe1 as an antiviral.

Minor issues:

The source of the CPBAe1 and ProCP proteins should be clarified for the ELISA with the DENV virion (Fig 4B).

Page 3

english needs to be clarified "...dengue virus, Zika virus and chikungunya as well as the malaria parasite.."

"exiting of viruses from the midgut or (3) impede" should read "exiting of viruses from the midgut cells or (3) impede"

Page 22 Site-directed mutagenesis needs more details including primers, kits used etc

Reviewer #2 (Comments to the Authors (Required)):

This work sought to analyze the structural basis for the impact of *Aedes aegypti* procarboxypeptidase B1 on Dengue virus replication in the mosquito midgut. Further, they characterized the interaction between procarboxypeptidase and DENV envelope (E) protein, virus-like particles, and infectious virions. They found that the residues Asp18A, Glu19A, Glu85, Arg87, Arg89 of PCPBAe1 are essential for its interaction with DENV. Such interaction between E protein and procarboxypeptidase could be well supported by their data based on crystal/structural analysis.

I only have several minor comments for this work:

1. This work pretty much focuses on the "interaction" between procarboxypeptidase B1 and E protein of DENV. Thus I am wondering the current title of this work is kind of too big (not so many for "Dengue virus replication"). Thus, can author try to make a more focus or straightforward title?
2. In page 9, it is better to move the paragraph "All the ELISA experiments below were done in triplicates. Data was analyzed by independent unpaired t-test between two sets of data (control and treated samples), and by two-way ANOVA for multiple groups of data using GraphPad Prism version 7.00. Results are shown as mean {plus minus} S.D. *p < 0.05, **p < 0.01 and ***p < 0.001, ****p < 0.0001." to figure legends, but not in the Results.
3. They did most of protein-virus interaction assay using ELISA. This is fine. However, there are might be artificial. Thus, they should at least discuss a little bit in Discussion if they can use more methods to examine the interactions between virus/E protein and enzyme. For example, they may use a competition assays of anti-E protein serum or antibody to validate the interactions.
4. The second paragraph of Discussion is too long. Can it be split to two-three paragraphs. Also, it is not necessary to repeat the most of results in Discussion section with repeatedly citing figure numbers (e.g. Figure 3A and 3C).

5. They found that "with the four DENV serotypes showed different levels of interaction in the following decreasing order: DENV-3 > DENV2 > DENV-4 > DENV-1". Since E protein sequence for four DENV serotypes are available, can authors discuss a little bit really sequence difference of E protein in these different serotypes of DENV contributed to these decreasing orders? One possible reasons may just because other non-specific interactions lead to such difference.
6. Another issue is that: as shown by Figure 7, PCPBAe1 may help to limit the life cycle of DENV in mosquito midgut. This may serve as an innate immunity strategy for mosquito? However, on the other hand, this may not be a good strategy for enhancing the replication and spreading of DENV and potentially other vector-born viruses? Can author discuss a little bit about the significance about this balance?
7. They discuss that "However, because this difference is only marginal (and not significant), these findings cannot wholly explain why PCPBAe1 only suppresses the release of DENV-2 in mosquito C6/36 cell line but not in a mammalian Vero cell line". This makes sense as our experience show that Vero cell line is actually not a good cell line as like C6/36 cells for supporting cultures of DENV.

We sincerely thank both reviewers for their constructive comments that greatly improved the manuscript. All the new changes in the revised manuscript are provided in “blue” font.

Reviewer #1

The essential roles of midgut metalloproteases (MCPs) have been previously established and PCPBAe1 has been shown to regulate dengue virus accumulation and release from midgut cells. Due to these crucial roles the Gavor et al have undertaken a structural and molecular characterisation of the full-length PCPBAe1 (proenzyme) and the mature peptidase domain (CPBAe1) and interactions with the dengue envelope (E) protein.

Gavor *et al* present the crystal structure of the PCPBAe1 to 2.08 Å resolution (PDB: 7EQX). The validation report has not been provided however a brief look over the data collection and refinement statistics shows that completeness is low in the high-resolution shell and perhaps the data should be cut at a slightly lower resolution (2.1 or 2.2 Å). An analysis of the interactions mediating pro binding is provided and specific interactions occluding access to the substrate binding region are noted. Substrate binding was explored by structural comparison and site directed mutagenesis to demonstrate the mechanism by which CPBAe1 accommodates Arg substrates.

We agree with this comment that the completeness of the last resolution bin is a bit low. Since we want to use all the reflections, we have used the maximum resolution of the data following the suggestions in the literature (Powell, 2021). Along with the revised manuscript we have submitted the validation report for the crystal structure of PCPBAe1 as a reference material.

Previous yeast-2-hybrid studies have established an interaction occurs between PCPBAe1 the dengue E protein/virion resulting in intracellular virus particle accumulation. Binding was measured by ELISA for combinations of insect (ICP) and E coli (ECP) produced PCPBAe1 and DENV2 E protein. ICP showed a higher interaction with all the viral antigens/virions as compared with ECP. Whilst the ICP and ECP proteins show similar molecular weights in the SDS-PAGE there is a lower molecular weight band in the ICP (Supp Fig 1C) sample, this sample should be investigated by mass-spectrometry to determine whether this is a contaminant or the mature CPBAe1 protein.

Thank you very much for pointing out this and suggesting a mass spectrometry identification. However, during this revision before conducting a mass spec experiment, we have again purified ICP, and this time ensured an extensive imidazole wash. Upon extensive imidazole wash (i.e., 30 mM imidazole concentration in wash buffer compared with 5 mM concentration previously), there is no impurity observed in the new SDS PAGE. This new SDS-PAGE gel picture is now used to replace the previous **Supplementary Fig. S1C**. Next, we have repeated the interaction studies with ICP and observed similar interaction profile with DENV - E protein, VLP and virion. As such, now the sample is pure, we have not conducted any mass spec experiments.

In the discussion the authors explain that the difference in glycan composition at Asn67 might explain the differences in suppression of DENV2 release from insect C6/36 and mammalian Vero cells. This is insignificant and as Gavor et al note "other conditions present in the mosquito cell line microenvironment....are likely involved in virus inhibition". Tham et

al (2014) also reported an increase in extracellular DENV2 within the Vero cell line when treated with PCPBAe1 and this would preclude the use of PCPBAe1 as an antiviral.

Thank you very much for bringing this point to our attention. Indeed, the entry mechanism of dengue virus into mosquito and human cells is different, and this is controlled by multiple factors including differences in putative receptors and attachment factors (Reyes-del Valle et al., 2014). For example, virus internalization might occur through distinct entry pathways, including clathrin-mediated or non-classical clathrin-independent endocytosis, depending on whether mosquito or human cell or specific host cells (Reyes-del Valle et al., 2014). This point has been added to the discussion section (Page 15, paragraph 2).

Further we understood that the reviewer thinks that although PCPBAe1 serves as an anti-dengue viral molecule in the mosquito, it may not represent an effective dengue or other mosquito-borne viral diseases control strategy as the inhibition of viral propagation seems to only occur in the vector. We have also added this information in the discussion section (last paragraph page numbers 16).

Minor issues

The source of the CPBAe1 and ProCP proteins should be clarified for the ELISA with the DENV virion (Fig 4B).

As suggested we have indicated in Fig. 4B legend that both CPBAe1 (mature peptidase domain of PCPBAe1) and the ProCP (Pro-region of PCPBAe1) were produced from the *E.coli* expression system.

Page 3

English needs to be clarified "...dengue virus, Zika virus and chikungunya as well as the malaria parasite..."
"exiting of viruses from the midgut or (3) impede" should read "exiting of viruses from the midgut cells or (3) impede"

As suggested the first line, page 3, first paragraph of the manuscript has been reworded as "Mosquito-borne viruses such as dengue virus, Zika virus and chikungunya virus as well as the malaria parasite constitute a major threat to public health".

Also, the statement in Page 14, second paragraph has been addressed by adding the word "cells" as suggested by the reviewer.

Page 22 Site-directed mutagenesis needs more details including primers, kits used etc. As suggested the following primers used for the site-directed mutagenesis have been provided as **Supplementary Tables S3 and S4**. All the PCR reactions were carried out using KAPABIOSYSTEMS HiFi PCR reaction Kits (Roche) (please refer to Materials and Methods, section site-directed mutagenesis, page 22).

Supplementary table S3. Site-directed mutagenesis of PCPBAe1 and oligonucleotides

Peptide	Mutation	Oligonucleotide (5'-3')
Asp ²⁵¹ Glu		GCGGCGGGTGGTAGCGAAGATTGGGCGTTCGCG CGCGAACGCCCAATCTTCGCTACCACCCGCCG

Ser ²³⁹ Gly		TACACCGTGGGTAGCGGTACCAACGTTCTGTAT ATACAGAACGTTGGTACCGCTACCCACGGTGTA
Tyr ^{4A} -Glu ^{21A}	Asp ^{18A} Glu ^{19A} del	GTTCCGGAAAGCCCCGGCGGAAATCCTGTAT ATACAGGATTTCCGCCGGGCTTTCCGGAAC
Ala ^{81A} -Glu ²⁰	Asp ⁵ Ala, Arg ⁶ Ala, Asp ⁷ Ala	GGCATGAGCATGCTGGCGGCGGGTGTAGCACCAGCTAC GTAGCTGGTGCTACCGCCGCCAGCATGCTCATGCC
Gly ¹⁶² -Arg ¹⁷²	Glu ¹⁶⁸ Ala, Glu ¹⁷⁰ Ala, Arg ¹⁷² Ala	GAAACCGCGTTTTAGCGCGCCGGCGACCGCGGGTGTGCGTGTGCG CGCATCACGCACCGCCGGTTCGCCGGCGGCTAAACCGGGTTTC
Val ³⁰⁰ -Phe ³⁰⁵ del	Val ³⁰⁰ -Phe ³⁰⁵ del	GCGATGGCGCTGAAATAAGTTGCGCAAATGTTT AAACATTTGCGCAACTTATTTTCAGCGCCATCGC
ProCP	ProCP	AGCATGCTGGACTAGCGTGATGTGAGC GCTCACATCACGCTAGTCCAGCATGCT

Supplementary table S4. Oligonucleotides for the generation of de-glycosylated DENV2 mutants

Primer Name	Oligonucleotide (5'-3')
D2Y98P-F1-FP	CTGGTTTAGTGAACCGTCAGAGTAGTTAGTCTACGTGGAC
D2Y98P-F1-RP	CTCACAACGCAACCCTATCGGCCTGCACCATAACTCC
D2Y98P-F2-FP	TGGGAGTTATGGTGCAGGCCGATAGTGGTTGCGTTGTG
D2Y98P-F2-RP	ATTGCTGGAAGGTATCTCTTTGTTTTTCTGCTCCTGG
D2Y98P-F3-FP	ACCCAGGAGCAGGAAAAACAAAGAGATACCTTCCAGCAATAGTCAG AGAAG
D2Y98P-F3-RP	TTTGAAGACGCACCAGATTCCAACCATATGTTGACATGG
D2Y98P-F4-FP	CCCATGTCAACATATGGTTGGAATCTGGTGCCTCTCAAAG
D2Y98P-F4-RP	TGGAGATGCCATGCCGACCCAGAACCTGTTGATTCAAC
Vector (CMV, HDV ribozyme and SV40 PA) FP	CTGTTGAATCAACAGGTCTGGGTCGGCATGGCATCTC
Vector (CMV, HDV ribozyme and SV40 PA) RP	GTCCACGTAGACTAACTACTCTGACGGTTCACTAAACCAGC
67-69 NTT-QTV FP	CAAAGCTGACCCAAACAGTTACAGCATCTCGCTGCCCAAC
67-69 NTT-QTV RP	CGAGATGCTGTAAGTGTGGGTCAGCTTTGCCTCTATAC
153 N-Q FP	TGCAGTCGGACAAGACACAGGAAAAC
153 N-Q RP	TCCGACTGCATTCTTCCCCTG

Reviewer 2

This work sought to analyze the structural basis for the impact of *Aedes aegypti* procarboxypeptidase B1 on Dengue virus replication in the mosquito midgut. Further, they characterized the interaction between procarboxypeptidase and DENV envelope (E) protein, virus-like particles, and infectious virions. They found that the residues Asp18A, Glu19A, Glu85, Arg87, Arg89 of PCPBAe1 are essential for its interaction with DENV. Such interaction between E protein and procarboxypeptidase could be well supported by their data based on crystal/structural analysis.

1. This work pretty much focuses on the "interaction" between procarboxypeptidase B1 and E protein of DENV. Thus, I am wondering the current title of this work is kind of too big (not so many for "Dengue virus replication"). Thus, can author try to make a more focus or straightforward title?

As suggested to make it more specific we have changed the title of the manuscript as "Structure of *Aedes aegypti* procarboxypeptidase B1 and its interactions with dengue virus for controlling infection"

2. In page 9, it is better to move the paragraph "All the ELISA experiments below were done in triplicates. Data was analyzed by independent unpaired t-test between two sets of data (control and treated samples), and by two-way ANOVA for multiple groups of data using 10 GraphPad Prism version 7.00. Results are shown as mean {plus minus} S.D. *p < 0.05, **p < 0.01 and ***p < 0.001, ****p < 0.0001." to figure legends, but not in the Results.

As suggested, we have moved the statement "All the ELISA experiments....." from Page 9 to the figure legend of the ELISA figures (Fig. 4 and Fig. 6).

3. They did most of protein-virus interaction assay using ELISA. This is fine. However, there might be artificial. Thus, they should at least discuss a little bit in Discussion if they can use more methods to examine the interactions between virus/E protein and enzyme. For example, they may use a competition assays of anti-E protein serum or antibody to validate the interactions.

As suggested, to clarify this point, we have added the following statement in the discussion section, Page 14, beginning of paragraph 2: "Furthermore, the HDXMS results suggest that PCPBAe1 binds across the E protein's dimeric interface by targeting flexible loop regions but not the conserved fusion loop (Asp⁹⁸-Lys¹¹⁰). The 4G2 anti DENV-E protein antibody used in our ELISA experiment targets the conserved fusion loop epitope, consistent with the fact that the antibody detects DENV E protein/virion/VLP even in the presence of PCPBAe1".

4. The second paragraph of Discussion is too long. Can it be split to two-three paragraphs? Also, it is not necessary to repeat the most of results in Discussion section with repeatedly citing figure numbers (e.g., Fig. 3A and 3C).

As suggested, the second paragraph of the discussion has been split into two paragraphs to convey specific message in each paragraph and make reading easier. Moreover, we try to minimize the repetition of the results section in the discussion section (i.e., we compressed

paragraph 2 of the discussion section) along with the removal of repeatedly cited figures from the discussion.

5. They found that "with the four DENV serotypes showed different levels of interaction in the following decreasing order: DENV-3 > DENV2 > DENV-4 > DENV-1". Since E protein sequence for four DENV serotypes are available, can authors discuss a little bit really sequence difference of E protein in these different serotypes of DENV contributed to these decreasing orders? One possible reason may just because other non-specific interactions lead to such difference.

We thank the reviewer for pointing this out. To address this comment, we closely examined the E protein interaction sites and have added the following paragraph in the discussion (Pages 14 and 15, Paragraph last and first paragraphs, respectively).

“There are three discontinuous interaction sites on the DENV E protein that bind to PCPBAe1. We systematically compared the sequences of these sites to understand the differential interactions (DENV-3 > DENV2 > DENV-4 > DENV-1) with DENV serotypes. At first, we compared the sites of DENV-3 and DENV-1, which are at the opposite ends of affinities. Site 3 is completely conserved between DENV-3 and DENV-1, while site 2 shows a single residue change (Thr²⁴²Asn) (Fig. 7). In contrast, site 1 shows several key differences. Specifically, the two hydrophilic Thr residues to hydrophobic residues in DENV-3 (Thr⁶⁸Ile and Thr⁸¹Val). These hydrophobic residues may enhance the interaction due to the tendency to bury themselves in the interaction surface. In addition, a Pro⁸³ residue replaces Val⁸³ in DENV-3, creating a proline bracket with Pro⁷⁸ residue(Kini & Evans, 1995). Such proline brackets are known to enhance protein-protein interactions(Manjunatha Kini & Evans, 1995). Thus, it appears that the Site 1 (62-82 residues) mostly defines the affinity differences observed between DENV-3 and DENV-1. There are several amino acid residue differences observed in all three sites between DENV-2 and DENV-4. Interestingly, sites 1 of both DENV-2 and DENV-4 have slightly shorter regions within the proline brackets compared to DENV-3. They also have a conserved Pro²⁴³ residue unlike DENV-1 and DENV-3. It is unclear which of these changes are responsible for the observed differences in affinity in these two serotypes”.

6. Another issue is that: as shown by Figure 7, PCPBAe1 may help to limit the life cycle of DENV in mosquito midgut. This may serve as an innate immunity strategy for mosquito? However, on the other hand, this may not be a good strategy for enhancing the replication and spreading of DENV and potentially other vector-borne viruses? Can author discuss a little bit about the significance about this balance?

We understand the reviewers point here to mean that even though PCPBAe1 serves as an anti-dengue viral molecule in the mosquito, it may not represent an effective dengue or other mosquito-borne viral diseases control strategy as the inhibition of viral propagation seems to only occur in the vector. We have added this information in the last paragraph of the discussion section, page numbers 16, last paragraph.

7. They discuss that "However, because this difference is only marginal (and not significant), these findings cannot wholly explain why PCPBAe1 only suppresses the release of DENV-2 in mosquito C6/36 cell line but not in a mammalian Vero cell line". This makes sense as our

experience show that Vero cell line is actually not a good cell line as like C6/36 cells for supporting cultures of DENV.

Thank you very much for this insight. We have added a brief statement “Furthermore, C6/36 is a better cell line for DENV infection than Vero cell line”, in the last line (Page 15).

October 7, 2021

RE: Life Science Alliance Manuscript #LSA-2021-01211R

Prof. J Sivaraman
National University of Singapore
Department of Biological Science
14 Science Drive 4
Singapore, Singapore 117543
Singapore

Dear Dr. Sivaraman,

Thank you for submitting your revised manuscript entitled "Structure of *Aedes aegypti* procarboxypeptidase B1 and its interactions with dengue virus". We would be happy to publish your paper in Life Science Alliance pending final revisions necessary to meet our formatting guidelines.

- please add ORCID ID for the corresponding author-you should have received instructions on how to do so
- please add the Twitter handle of your host institute/organization as well as your own or/and one of the authors in our system
- please note that titles in the system and manuscript file must match
- please add your main, supplementary figure, and table legends to the main manuscript text after the references section
- please add callouts for Figures 5C, 6A, 8A and B, S1B and E-H, S3A-C to your main manuscript text;

Figure checks:

- there is a vertical splice in Figure S1C, between the first and the second column. Please provide source data for this figure as a separate file.

A. FINAL FILES:

B. MANUSCRIPT ORGANIZATION AND FORMATTING:

Sincerely,

October 19, 2021

RE: Life Science Alliance Manuscript #LSA-2021-01211RR

Prof. J Sivaraman
National University of Singapore
Department of Biological Science
14 Science Drive 4
Singapore, Singapore 117543
Singapore

Dear Dr. Sivaraman,

Thank you for submitting your Research Article entitled "Structure of Ae. aegypti procarboxypeptidase B1 and its binding with DENV for controlling infection". It is a pleasure to let you know that your manuscript is now accepted for publication in Life Science Alliance. Congratulations on this interesting work.

DISTRIBUTION OF MATERIALS:

Again, congratulations on a very nice paper. I hope you found the review process to be constructive and are pleased with how the manuscript was handled editorially. We look forward to future exciting submissions from your lab.

Sincerely,
